# Changes in Sphingolipid Profile of Benzo[a]pyrene-Transformed Human Bronchial Epithelial Cells Are Reflected in the Altered Composition of Sphingolipids in Their Exosomes

**DOI:** 10.3390/ijms22179195

**Published:** 2021-08-25

**Authors:** Miroslav Machala, Josef Slavík, Ondrej Kováč, Jiřina Procházková, Kateřina Pěnčíková, Martina Pařenicová, Nicol Straková, Jan Kotouček, Pavel Kulich, Steen Mollerup, Jan Vondráček, Martina Hýžďalová

**Affiliations:** 1Department of Pharmacology and Toxicology, Veterinary Research Institute, Hudcova 296/70, 62100 Brno, Czech Republic; slavik@vri.cz (J.S.); kovac@vri.cz (O.K.); prochazkova@vri.cz (J.P.); pencikova@vri.cz (K.P.); parenicova@vri.cz (M.P.); strakova@vri.cz (N.S.); kotoucek@vri.cz (J.K.); kulich@vri.cz (P.K.); 2Section for Occupational Toxicology, National Institute of Occupational Health, 0304 Oslo, Norway; steen.mollerup@stami.no; 3Department of Cytokinetics, Institute of Biophysics of the Czech Academy of Sciences, Kralovopolska 135, 61265 Brno, Czech Republic; vondracek@ibp.cz

**Keywords:** sphingolipid, glycosphingolipid, eicosanoids, exosomes, in vitro cell transformation

## Abstract

Sphingolipids (SLs), glycosphingolipids (GSLs), and eicosanoids are bioactive lipids, which play important roles in the etiology of various diseases, including cancer. However, their content and roles in cancer cells, and in particular in the exosomes derived from tumor cells, remain insufficiently characterized. In this study, we evaluated alterations of SL and GSL levels in transformed cells and their exosomes, using comparative HPLC-MS/MS analysis of parental human bronchial epithelial cells HBEC-12KT and their derivative, benzo[a]pyrene-transformed HBEC-12KT-B1 cells with the acquired mesenchymal phenotype. We examined in parallel SL/GSL contents in the exosomes released from both cell lines. We found significant alterations of the SL/GSL profile in the transformed cell line, which corresponded well with alterations of the SL/GSL profile in exosomes derived from these cells. This suggested that a majority of SLs and GSLs were transported by exosomes in the same relative pattern as in the cells of origin. The only exceptions included decreased contents of sphingosin, sphingosin-1-phosphate, and lactosylceramide in exosomes derived from the transformed cells, as compared with the exosomes derived from the parental cell line. Importantly, we found increased levels of ceramide phosphate, globoside Gb3, and ganglioside GD3 in the exosomes derived from the transformed cells. These positive modulators of epithelial–mesenchymal transition and other pro-carcinogenic processes might thus also contribute to cancer progression in recipient cells. In addition, the transformed HBEC-12KT-B1 cells also produced increased amounts of eicosanoids, in particular prostaglandin E2. Taken together, the exosomes derived from the transformed cells with specifically upregulated SL and GSL species, and increased levels of eicosanoids, might contribute to changes within the cancer microenvironment and in recipient cells, which could in turn participate in cancer development. Future studies should address specific roles of individual SL and GSL species identified in the present study.

## 1. Introduction

Mammalian cells are known to secrete heterogeneous extracellular vesicles (EVs) that contain nucleic acids, enzymes, and various metabolites, including lipids. Owing to their rich composition and capacity to interact with other recipient cells, EVs have been shown to play functional roles in multiple physiological and pathological processes, both directly in the microenvironment of the EV-releasing cells, and in distant organs, thus constituting an important type of intercellular autocrine, paracrine, and endocrine communication, as well as compound exchange [1,2,3]. Importantly, neoplastic cells have been shown to release EVs, which may modify the phenotype of host cells, in order to facilitate tumor growth. EVs can also be readily exchanged within cancer cell populations, which can further promote cell proliferation [4,5,6], epithelial–mesenchymal transition (EMT), and migration or invasion of tumor cells [7,8,9]. These processes have all been covered in several recent reviews [10,11,12].

Exosomes are small EVs generated within endosomal compartments of the cells (with sizes up to 100–200 nm). Their bioactivity is associated not only with their RNA and protein content, but it can be also linked with specific lipids constituting exosomes. Indeed, they have been shown to transport numerous bioactive lipids, as shown previously for eicosanoids, fatty acids, cholesterol, lysophosphatidylcholine, and sphingomyelin (SM), as well as the enzymes involved in lipid metabolism, including phospholipases D and A2 [13,14]. Therefore, exosomes may act as intercellular transporters of lipid molecules and related enzymes, which are involved in various pathophysiologies, including cancer [15], and that both lipids and lipid metabolic enzymes transported by exosomes may alter the status of recipient cells.

Previously, several studies have reported the lipid composition of exosomes released from various types of cells [13,16,17,18,19]; however, the current knowledge about the molecular lipid composition is far from being complete. In a pivotal study, the enrichment of sphingolipids (SLs) and glycosphingolipids (GSLs) has been observed in the exosomes released from the prostate carcinoma PC-3 cell line [16]. Since exosomes originate from endosomes, where SLs are internalized, and several SLs have been reported as being essential for a proper exosomal biogenesis [3], significant amounts of SLs are present in exosomes. SLs can modulate cell growth, proliferation, survival, senescence, apoptosis, as well as cell migration and dissemination during cancer progression [20,21]. Similar to SLs, various GSLs have been reported to be involved in cancer development, playing multiple roles in cell proliferation, EMT, or in the control of migration and invasion of tumor cells [22,23]. Importantly, SL and GSL composition appears to be frequently altered in cancer cells, which is linked with differential expression of the genes involved in SL/GSL metabolism [24,25,26]. Additional lipid signaling molecules identified in exosomes, including eicosanoids, such as prostaglandin E2 (PGE_2_), have also been reported to play significant roles in the control of inflammation, neoangiogenesis, responses of immune cells towards tumor cells, or in cancer progression. Their production and release from the cancer cells can also be changed, as compared with parental non-transformed cells [27]. A direct link has been identified between aberrant levels of phosphorylated SLs and activation of cytosolic phospholipase A2α and induction of arachidonic production, which is the rate-limiting step in eicosanoid synthesis, or activation of cyclooxygenase-2, as the first enzyme of prostaglandin biosynthesis [28,29].

In a previous study [30], phenotypical transformation of human bronchial epithelial cells HBEC-12KT induced by a chronic exposure to benzo[a]pyrene (BaP) has been reported. Here, in this follow-up study using the same cellular model, we focused on the composition of SLs and GSLs in both non-transformed (HBEC-12KT) and BaP-transformed (HBEC-12KT-B1) cells. To the best of our knowledge, this is the first report directly comparing the SL/GSL profiles of human epithelial cells with their fully transformed mesenchymal derivatives, combining it with characterization of the SL/GSL profiles of exosomes released from non-transformed and carcinogen-transformed cells. In parallel, we performed targeted gene expression analysis focused on GSL metabolism in these cell models. Finally, we evaluated the concentrations of eicosanoids released from both normal and transformed cell lines, because deregulation of SL metabolism can also be linked with altered metabolism of arachidonic acid, as described above [29]. Importantly, regardless of several exceptions, the changes in the SL/GSL concentrations in exosomes and eicosanoid release matched the differences in the expression of the genes linked with SL/GSL and prostaglandin metabolism well, and they reflected the content of SL/GSLs in normal or transformed cell lines. Cell transformation via a well-defined chemical carcinogen was identified as a major factor deregulating SL, GSL, and eicosanoid metabolism, which may lead to the release of prostaglandins and exosomes with altered SL/GSL profiles, which might in turn contribute to the processes associated with cancer progression in recipient cells.

## 2. Results

### 2.1. Characterization of Isolated Exosomes

Exosomes released from HBEC-12KT and HBEC-12KT-B1 cells, maintained in conditioned medium, were isolated by differential ultracentrifugation and characterized by the multi-angled dynamic light scattering (MADLS^®^) technique. We determined the hydrodynamic size, polydispersity index (PdI), and concentration (Table 1). Samples generally showed a high degree of homogeneity with a low PdI value of 0.130 and 0.123, and with an average hydrodynamic size of 158 and 162 nm in diameter for HBEC-12KT and HBEC-12KT-B1, respectively. The overall homogeneity of the samples was also apparent from the results of distribution analysis (Figure 1A). The numbers of exosomes were 2.53 × 10^8^ (for HBEC-12KT) and 2.26 × 10^8^ (for HBEC-12KT-B1) exosomes per 1 × 10^6^ cells.

The quality of exosome isolation was further verified by detection of the exosome-related marker CD63/LAMP-3, using transmission electron microscopy (using a combination of anti-CD63 antibody/Protein A-colloid gold labelled secondary antibody) and flow cytometry (using phycoerythrin (PE)-conjugated anti-CD63 antibody), respectively (Figure 1B,C).

### 2.2. HPLC-MS/MS and Flow Cytometric Analyses of SL and GSL Profiles in Transformed and Parental HBEC-12KT Cells, and in Their Exosomes

The major aim of this study was to evaluate the levels of SLs and GSLs in the BaP-transformed mesenchymal HBEC-12KT-B1 cells vs. the non-transformed parental epithelial HBEC-12KT cells, and in exosomes released from both cell lines. Comparative HPLC-MS/MS analysis revealed significant differences between the concentrations of SLs and GSLs in parental human epithelial cells and in the transformed mesenchymal counterparts, which were reflected also in the SL/GSL profiles in the respective exosomes (Figure 2). In the HBEC-12KT-B1 cells, we found significantly decreased total levels of sphinganine (dhSph), sphinganine phosphate (dhS1P), dihydroceramides (dhCer), ceramides (Cer), dihydrosphingomyelin (dhSM), and sphingomyelin (SM). On the other hand, elevated levels of ceramide phosphate (CerP), sphingosin (Sph), and sphingosin-1-phosphate (S1P) were determined in the transformed HBEC-12KT-B1 cells, as compared with their content in non-transformed HBEC-12KT cells (Figure 2A). With regard to GSLs, the levels of HexCer (represented mostly by GlcCer; data not shown), sulfatide (a product of GalCer pathway), and complex GSL GM3 were significantly lower in HBEC-12KT-B1 cells. In contrast, the synthesis of LacCer, Gb3, and especially GD3 appeared to be strongly upregulated in the HBEC-12KT-B1 cells (Figure 2B).

We also evaluated the content of SLs and GSLs in the exosomes isolated from transformed and non-transformed cells. In general, the levels of SL and GSL species in the exosomes released from the transformed HBEC-12KT-B1 cells exhibited a similar profile as in the HBEC-12KT-B1 cells (Figure 2A,B). Nevertheless, there were a few differences; although the levels of both Sph and LacCer were higher in transformed cells than in the parental cell line, a decrease of Sph and LacCer was observed in exosomes isolated from the transformed cells, and no difference was found in S1P levels, which were also increased in the transformed cells.

More detailed information about the changes in individual SL and GSL species, both in cells and in exosomes, is provided in the Appendix A Appendix A. While a few minor species, such as Cer C20:0, SM C18:0, SM C20:0, and Glc C18:0, showed different trends than the ones calculated for the total levels of SL/GSLs, a majority of individual SL/GSL species exhibited similar profiles to the total SL/GSLs. This also included the abundant C16:0, C24:0, and C24:1 species.

Further, we determined and compared the levels of selected GSLs on the cellular surface using flow cytometry. Whereas the surface marker SSEA4 (the final metabolite of Gb3 glycosphingolipid) was not detected at significant levels, increased levels of LacCer and GD2 (the metabolite of GD3) were found in BaP-transformed cells (Figure 3).

### 2.3. HPLC-MS/MS Analysis of Eicosanoids Released from Transformed and Parental HBEC-12KT Cells

Prostaglandins and other eicosanoids represent another group of lipid signaling molecules involved in cancer progression, cardiovascular diseases, and other pathological processes [31,32]. Induction of CerP and S1P synthesis might also be involved in induction of eicosanoid metabolism [28,29], thus linking SLs and eicosanoid production. Therefore, we further extended our comparative study and included HPLC-MS/MS analysis of eicosanoids released from the transformed HBEC-12KT-B1 cells with a mesenchymal phenotype, and from non-transformed bronchial epithelial HBEC-12KT cells (Figure 4).

Levels of the primary precursor of bioactive eicosanoids, arachidonic acid (AA), was significantly higher in HBEC-12KT-B1 cells, together with a higher generation of hydroxyoctadecadienoic acids 13-HODE and 9-HODE, which, together with increased levels 8-iso-prostaglandin 2alpha (8-iso-PGF_2α_), could be indicative of increased oxidative stress, accompanied by excessive lipid peroxidation [33,34]. Importantly, significantly higher levels of a series of prostaglandins, including PGD_2_, PGJ_2_, PGA_2_, PGF_2α_, and especially more than a 6000-fold increase in the PGE_2_ concentration, were found in the growth medium of mesenchymal HBEC-12KT-B1 cells, as compared with the production of prostaglandins in non-transformed HBEC-12KT cells (Figure 4B).

### 2.4. Expression Analysis of Genes Participating in the GSL and Eicosanoid Metabolism

Based on the results obtained from HPLC-MS/MS analysis, we further examined the expression of genes driving the metabolism of sphingolipids with a particular focus on HexCer, LacCer, and complex GSLs (Figure 5A,B), as well as on the enzymes producing selected prostaglandins (Figure 5C). Here, we did not evaluate the mRNA levels of the genes catalyzing ceramide biosynthesis, as there is already extensive information available about the deregulation of this pathway in cancer cells [20,35,36]. The mRNA levels of GlcCer synthase UGCG, GalCer synthase UGT8, and LacCer synthase B4GALT5, as well as the specific GlcCer transport protein FAPP2 and GSL hydrolases GBA, GBA2, and GLA, were not significantly changed in the transformed HBEC-12KT-B1 cells. On the other hand, galactosylceramidase GALC, beta-galactosidase GLB1 (involved, e.g., in conversion of LacCer to GlcCer), LacCer synthase B4GALT6, and ceramide transporter CERT (which is a limiting protein of SM synthesis) were all downregulated in the transformed mesenchymal cells (Figure 5A). This did not correspond with the changes of their respective products in the BaP-transformed cells, GlcCer and LacCer.

On the other hand, increased levels of Gb3 synthase A4GALT and GD3 synthases ST8SIA1, ST8SIA4, and ST8SIA5 (Figure 5B) corresponded well with increased levels of globoside Gb3 and ganglioside GD3, respectively. Similarly, the reduced levels of GM3 synthase ST3GALT5, GM2 synthase B4GALNT1, and GM1a synthase B3GALT4 were all in good accordance with significantly lower levels of gangliosides GM3, GM2, and GM1a in the BaP-transformed cells, as compared with their content in non-transformed epithelial HBEC-12KT cells (Figure 5B).

The increased concentrations of several eicosanoids released from the transformed mesenchymal cells into cell culture media led us to further examine the expression of several key genes of prostaglandin metabolism. However, we found that only PGD_2_ synthase PTGDS and PGE_2_ synthase PTGES2 were increased in the transformed cells. Cyclooxygenase-2 (PTGS2), as well as the major inducible PGE_2_ synthase PTGES, were both downregulated in the BaP-transformed cells (Figure 5C).

## 3. Discussion

Neoplastic cells produce exosomes (and larger EVs), which may modify the phenotype of recipient cells [10,11,12]. For example, EVs released by mesenchymal-like prostate carcinoma cells modulated the EMT status of recipient epithelial-like carcinoma cells [9]. Therefore, changes in the exosomal content of RNA and protein components of exosomes reflecting changes occurring during cell transformation and EMT have been intensively studied, as they may affect the recipient cells. However, far less is known about the SL/GSL composition of non-transformed and transformed cells, in particular with regard to the SL/GSL composition of their exosomes.

Numerous lipid species, including sphingolipids and eicosanoids, have been shown to play important roles in carcinogenesis, both as lipid signaling molecules and as modulators of the structure/functions of cell membranes [20,31,32]. This study aimed: (1) to compare the SL and GSL content in the human bronchial epithelial cell line HBEC-12KT and in their BaP-transformed counterpart, HBEC-12KT-B1 cells with acquired mesenchymal status [30], in order to identify possible links between SL/GSL concentrations and the expression of genes related to SL/GSL metabolism; (2) to compare SL/GSL composition in exosomes released from both cell lines and to explore possible relationships between the SL/GSL profiles in exosomes and the mesenchymal epithelial cell phenotype, respectively. These changes were also compared with the concentrations of eicosanoids released from both cell lines. We hypothesized that the lipid content of exosomes (and growth medium generally) would largely depend on the lipid profiles of the cells serving as a source of exosomes [18].

Interestingly, we found that the total levels of SLs serving as constituents of the ceramide biosynthetic pathway, including Cer and SM, were significantly lower in the transformed cells with the mesenchymal phenotype. In contrast, significantly increased concentrations of Sph, S1P, and CerP were found in the transformed cells, when compared with their contents in normal epithelial cells. A number of genes/enzymes involved in SL metabolism have been reported to be deregulated in human cancer cells, and many of them seem to be linked with an increased S1P/Cer ratio, which is in turn associated with increased cancer cell survival, proliferation, and cancer progression [20]; in this study, the expression of genes related to SL metabolism was therefore not investigated further. The SL profile of exosomes mostly reflected the relative contents of SLs in the cells of their origin, with exception of Sph, S1P, and LacCer. Levels of Sph and LacCer were significantly lower in the exosomes isolated from the transformed cells, whereas the S1P concentration was similar in both types of exosomes. Nevertheless, because of the reduced levels of Cer in the transformed cells, a similar S1P/Cer ratio was found both in the exosomes and in the transformed mesenchymal cells, where they derive from. Importantly, exosomes from the transformed cells had a significantly higher amount of CerP, which is another SL signal molecule linked to cancer progression [20].

Induction of both CerP and S1P pathways has also been implicated in the activation of arachidonic acid metabolism, including increased PGE_2_ production [28,29]. Therefore, HPLC-MS/MS analysis of eicosanoids was performed in our study, in order to compare the release of these lipid signaling molecules from epithelial (parental) and transformed cells with the mesenchymal phenotype. Although the production of several eicosanoids with anti-inflammatory and pro-resolution activities, such as lipoxin LXA4 [32], PGD2, and PGJ2 [37], was increased in the transformed cells, a massive increase in prostaglandins and HETEs with pro-carcinogenic and pro-inflammatory effects, especially PGE_2_ [27], was observed. For example, 12-HETE and 20-HETE have been implicated in cell proliferation, inhibition of apoptosis, angiogenesis, and invasion/metastasis [38]. Additionally, oxidative stress markers derived from AA and linoleic acid metabolism, 9-HODE, 13-HODE, and 8-iso-PGF2alpha [33,34], were also increased in the BaP-transformed cells. This indicates that the transformed cells with the mesenchymal phenotype may release large quantities of eicosanoids, which could further contribute to cancer progression in recipient cells. However, a direct connection of these changes in the production of eicosanoids with differences in the gene expression of the respective enzymes was not identified. Nevertheless, it seems that activation of cytosolic PLA2α and PTGS2/COX-2 and not changes in their expression plays a major role in induction of eicosanoid metabolism. This is in accordance with previous studies showing that an increase in CerP and S1P and decrease in SM intracellular concentrations (which was the altered SL pattern found in mesenchymal HBEC-12KT-B1 cells in our study) have been implicated in the induction of eicosanoid metabolism [28,39].

The transformation of cells is often accompanied with a switch from the epithelial to the mesenchymal-like phenotype, which involves various processes contributing to EMT. A direct comparison of the lipidome of epithelial and mesenchymal cells after induction of EMT has been performed in MDCK epithelial cells, using a combination of TLC and shotgun MS analyses. Suppression of sulfatide has been found to be the most prominent effect of the EMT process, and very-long-chain fatty acids containing GM3 have also been suppressed during EMT; Cer, HexCer, and total GM3 levels were not affected [40]. However, other GSLs have not been investigated by these analytical techniques. Similar to this study, we found a reduced production of sulfatide and GalCer, both directly in HBEC-12KT-B1 cells and in their exosomes. Nevertheless, the expression of synthetic genes UGT8 and GAL3ST1 was not changed and galactocerebrosidase GALC was suppressed in the transformed cells. Thus, a direct link between changes in the expression of principal enzymes and the levels of GalCer and sulfatide was not identified. Recently, downregulation of the GALC gene has been observed in lung and head and neck cancer cells, which can be mediated via hypermethylation of its promoter [41]. Altogether, the exact role(s) of this GSL pathway in EMT and cancer progression deserves further attention.

The EMT process has also been reported to be associated with further GSL reprogramming, in particular the switch from synthesis of asialo-series of GSLs (e.g., GA2) to the synthesis of gangliosides (GM3, GD3). Induced expression of GM3 synthase ST3GALT5 and GD3 synthase ST8SIA1 as well as suppressed expression of GM1a synthase B3GALT4 has been proposed to contribute to EMT [26]. In our study, we used human HBEC cells transformed with the chemical carcinogen—BaP. The cell transformation of HBEC-12KT has led to both morphological and functional changes, including enhanced migration, the ability to form colonies in soft agar, as well as induction of the mesenchymal phenotype reflecting EMT [30]. In HBEC-12KT-B1 cells, we found reduced expression of ST3GALT5. In addition, GM2 synthase B4GALNT1 and B3GALT4 were downregulated, while GD3 synthases (ST8SIA1, 4, and 5) were upregulated; this was accompanied by a decrease of GM3, GM2, and GM1a levels and an increase of GD3 and GD2. This GSL pattern of the transformed cells was also reflected in a higher content of GD3 and reduced (albeit not significantly) levels of GM3 and GM1a in the exosomes isolated from the transformed cells.

The ratio of these gangliosides (i.e., GD3 vs. GM3 + GM2 + GM1a) was in good accordance with previous findings, reporting the same pattern and potential roles of cancer-associated GSLs [22]. GD3 (and GD2) can enhance the malignant properties of cancer cells, including increased cancer cell proliferation and dissemination, while monosialyl gangliosides GM3, GM2, and GM1a often suppress the malignant properties of cancer cells. These can be linked with the ability of GSLs to interact with cell surface receptors and adhesion molecules (mostly in the membrane lipid rafts) and to modify both intracellular and cell-to-cell signaling, as well as the interaction of cells with the extracellular matrix [22]. Importantly, a similar profile of gangliosides was also found in the exosomes released from the transformed HBEC-12KT-B1 cells.

Several studies demonstrated that globoside Gb3 and its synthetase A4GALT might be involved in EMT of primary epithelial cancer cells [23]. Levels of Gb3 have been found to be high in metastatic colon cancer and colon cancer cell lines bearing a highly invasive phenotype [42]. Further, increased cell surface Gb3 expression may lead to acquisition of cisplatin resistance in non-small lung cancer cells [43]. In our study, high levels of Gb3 were identified in the transformed bronchial cells as well as in their exosomes. This high content of Gb3 in the BaP-transformed cells can thus be linked to cell invasiveness and/or drug resistance in recipient cells; therefore, its high content in the exosomes derived from the transformed cells may also affect these endpoints in recipient cells. This suggests that more attention should be paid to Gb3 expression in future studies.

In conclusion, the results (summarized graphically in Figure 6) presented here indicate that a majority of SLs (dhSph, dhCer, dhSM, Cer, SM, and CerP) and GSLs (GlcCer, sulfatide, Gb3, GD3 as well as GM3 and GM1a) are transported by exosomes originating from the transformed mesenchymal-like cells in the same relative pattern as in the cells of origin. Therefore, exosomes from the transformed cells carry a very similar SL/GSL profile to the cells alone. In particular, specific S1P/Cer and GD3/GM3 ratios, favoring carcinogenesis and cancer progression, are preserved in exosomes. The high content of CerP, GD3, and Gb3; altered S1P/Cer and GD3/GM3 ratios; and the release of PGE_2_ may all contribute significantly to induction of cell proliferation, migration, and invasion, as well as to pro-inflammatory effects in recipient cells. The results obtained in this study should be further explored in future experiments, including the mechanism(s) linked to the relatively lower content of Sph, S1P, and LacCer found in exosomes derived from transformed cells.

## 4. Materials and Methods

### 4.1. Cell Culture

Human normal bronchial epithelial HBEC-12KT cells (kindly provided by Prof. J.D. Minna [44]) were grown in LHC-9 medium (Gibco, Thermo Fisher Scientific, Waltham, MA, USA). The HBEC-12KT-B1 cell line has been generated by continuous exposure of HBEC-12KT cells to BaP (1 µM) for 12 weeks [28] and it was grown in LHC-9 medium supplemented with 10% fetal bovine serum (FBS; Sigma-Aldrich, Prague, Czech Republic). Cells were cultivated in collagen-coated (PureCol^®^, Advanced BioMatrix, Carlsbad, CA, USA) plastic cell-culture dishes (TPP, Transadigen, Switzerland) at 37 °C in an atmosphere of 5% CO_2_.

### 4.2. Isolation of Exosomes from Cell Conditioned Media

The HBEC-12KT and HBEC-12KT-B1 cells were seeded in 150 mm cell-culture dishes (TPP) and allowed to grow to 80% confluence. The growth medium was then replaced by 20 mL of exosome-free medium (filtered LHC-9 medium without FBS) per dish. After 48 h, the conditioned medium was collected and used for exosome isolation. Separation of exosomes was performed based on a slightly modified protocol of Théry et al. [45]. A series of centrifugation and ultracentrifugation steps (using Beckman Optima XPN-100, Beckman Coulter, Indianapolis, IN, USA) and a final purification through the 30% sucrose cushion were used according to Pospíchalová et al. (2015) [46]. The final pellets of exosomes were resuspended in 50–100 μL of filtered (0.22 μm PVDF filter) PBS. The exosomes were directly characterized and used for further analyses or stored at −80 °C until further processing.

### 4.3. Size and Concentration Characterization of Isolated Exosomes

Approximately 50 µL of the exosomal suspension were placed in a low volume quartz batch cuvette ZEN2112 (Malvern Pananalytical Ltd., Malvern, UK) and measured using the multi-angled dynamic light scattering technique (MADLS^®^) on Zetasizer Ultra (Malvern Pananalytical Ltd.) at a constant temperature of 25 °C. The device was equipped with a HeNe Laser (633 nm) and three detectors at the following angles: 173° (backscatter), 90° (side scatter), and 13° (forward scatter). The measured data were evaluated using ZS Xplorer software version 1.50 (Malvern Pananalytical Ltd.). The measured values included hydrodynamic size, polydispersity index (PdI), and concentrations of exosomes.

### 4.4. Transmission Electron Microscopy and Immunogold Labeling of Isolated Exosomes

In total, 50 µL of the exosomes suspension was incubated with the primary anti-human CD63 (MEM-259) antibody (Exbio, Prague, Czech Republic) at 4 °C overnight. Then, Protein A-labeled secondary antibody (Protein A-20nm Colloidal Gold Labeled Antibody, Sigma-Aldrich) was added to the suspension, incubated for 20 min at 20 °C, and then overnight at 4 °C with the secondary antibody. The labeled exosomes were suspended within a drop of MilliQ H_2_O. The resulting suspension was applied to a 300-Old mesh coated by formvar film and carbonated (Agar Scientific, Essex, UK). After drying, 2% of ammonium molybdate (Serva, Heidelberg, Germany) was placed onto the grid and excess was dried. Exosomes were then examined using a transmission electron microscope Philips 208 S Morgagni (FEI, Brno, Czech Republic) at 18,000–180,000× magnification and accelerating voltage of 80 kV.

### 4.5. Flow Cytometry

Isolated exosomes were incubated with PE-conjugated specific CD63 antibody (Anti-Hu CD63 PE (MEM-259) antibody, Exbio) for two hours at room temperature in the dark. The labeled exosomes were then analyzed with Amnis CellStream^®^ (Luminex Corporation, Austin, TX, USA), a flow cytometer with high sensitivity for analysis of small particles, and the data were evaluated by CellStream Analysis 1.2.55 software. Filtered (0.1 μm PVDF filter) PBS, unstained exosomes, and CD63 antibody without exosomes were used as the negative controls.

For analysis of the SSEA4, GD2, and LacCer levels on the surface of cells, the HBEC-12KT and HBEC-12KT-B1 cells were collected by accutase (Biosera, Nuaille, France) and surface GSL markers were stained with anti-human SSEA4 PE-CF594 (BD Horizon, Piscataway, NJ, USA), anti-human GD2 (Biolegend, San Diego, CA, USA), or anti-CDw17 (ThermoFisher Scientific). The anti-CDw17 antibody was used for staining and detection of LacCer. The mouse anti-human GD2 samples were stained with anti-mouse Alexa Fluor 488 (ThermoFisher Scientific) secondary IgG antibody, mouse anti-human CDw17 with anti-mouse IgG/IgM Alexa Fluor 488 (ThermoFisher Scientific) secondary antibody, according to the manufacturer’s protocols. PC3 DR cells and HMLE-EMT (both kindly provided by Dr. Karel Souček, Institute of Biophysics, Brno, Czech Republic) were used as positive controls for SSEA4 and GD2, respectively. The negative controls were stained with the secondary antibody or isotype control only. Finally, the cells were analyzed using an Amnis CellStream flow cytometer and the data were evaluated by CellStream Analysis 1.2.55 software.

### 4.6. Sphingolipid Extraction and Analysis

Extractions and analyses of SL/GSL levels were performed as previously described [47]. For lipid extraction, cells or isolated exosomes were harvested to glass tubes (13 × 100 mm) with 1.5 mL of methanol (Honeywell, Regen, Germany). The solution was then homogenized with sonic probe (about 20 strong strikes) and 50 µL of sample solution were taken for protein quantification. The protein content was measured in cell samples using bicinchoninic acid assay. After sonication, 0.75 mL of chloroform (Honeywell) were added, solution was again shortly homogenized with sonic probe, and after overnight extraction in room temperature, samples were placed in a heating box, evaporated under gentle flow of N_2_, and then reconstituted in 400 µL of methanol/chloroform 1:1 mixture. Samples were then transferred into vials and analyzed by HPLC-MS/MS.

Sphingolipid species (with exception of GSLs) were separated by reversed-phase HPLC (Dionex Ultimate 3000, Thermo Fisher Scientific) on a Gemini column C18, 5 µm, 250 × 4.6 mm (Phenomenex, Torrance, CA, USA) using a flow rate of 0.7 mL/min and gradient mobile phases RA/RB (RA = methanol/water 60/40, RB = methanol), starting at ratio RA/RB 60/40 for a total time of 69 min; formic acid and ammonium formate were used as eluent additives. GlcCer and GalCer were separated by normal phase HPLC (Dionex Ultimate 3000) on a Spherisorb column 5 µm Silica, 2.1 × 250 mm (Waters Chromatography, Dublin, Ireland) using a flow rate of 0.3 mL/min and gradient mobile phases NA/NB (NA = acetonitrile/methanol 99/1, NB = acetonitrile/methanol/water 40/47/13) for a total time of 48 min; formic acid and ammonium formate were used as eluent additives. GSLs were separated by reverse-phase HPLC (Dionex Ultimate 3000) on a ZORBAX Eclipse C8 4.6 × 250 mm column (Agilent Technologies, Santa Clara, CA, USA), using a flow rate of 0.8 mL/min gradient mobile phase RA/RB (RA= water/methanol 90/10, RB = methanol/isopropanol 1/1), starting at ratio RA/RB 61/39, for a total analysis time of 60 min; ammonium formate was used as the eluent additive. The tandem mass spectrometer (Hybrid Triplequad QTRAP 4500, AB Sciex, Concord, ON, Canada) was operated under following conditions: ESI in positive mode, drying air, collision energy, and fragmentor voltage were optimized for each species, and the spectrometer was run in multiple reaction monitoring (MRM) scan mode. MS standards used for quantification were purchased from Avanti Polar Lipids (Alabaster, AL, USA), and they are listed in the Appendix A Appendix A.

### 4.7. Liquid Chromatography Separation and Tandem Mass Spectrometry Conditions for Eicosanoid Analysis

Solid-phase extraction was used for extraction and purification of arachidonic acid metabolites from cell culture medium. SELECT HLB SPE 1 mL (30 mg) cartridges (Supelco, Prague, Czech Republic) were washed with 2 mL of methanol and 1 mL of 1% acetic acid. Then, 3 mL of medium with 0.1% of acetic acid were loaded onto the SPE column. The columns were then washed with 2 mL of 1% acetic acid. The cartridges were air-dried for 3 min with vacuum and analytes were eluted with 1.4 mL of methanol with 1% of acetic acid. Samples were dried under a stream of nitrogen, re-dissolved in 60 µL of methanol, and an aliquot of 5 µL was injected into the HPLC column.

Sample analyses were performed by using HPLC-MS/MS. An Agilent 1200 chromatographic system (Agilent Technologies, Waldbronn, Germany) was used, consisting of a binary pump, vacuum degasser, autosampler, and thermostatted column compartment. Separation of eicosanoids was carried out using an Ascentis Express C18, 2.1 × 150 mm, 2.7 µm particle size column (Supelco, Bellefonte, PA, USA) with a 25-min linear gradient from 30 to 100% of acetonitrile. The mobile phase contained 0.1% of formic acid. The flow rate of the mobile phase was 0.3 mL/min and the column temperature was set at 45 °C. A triple quadrupole mass spectrometer Agilent 6410 Triple Quad LC/MS (Agilent Technologies, Santa Clara, CA, USA) with an electrospray interface (ESI) was used for the detection of the analytes. The mass spectrometer was operated in the negative ion mode. Selected ion monitoring (SIM) at *m*/*z* 303.2 was used for quantification of arachidonic acid and multiple reaction monitoring (MRM) for all other analytes.

External standards for quantification of each eicosanoid were purchased from Cayman Chemical Company (Ann Arbor, MI, USA), and they are listed in the Appendix A Appendix A; formic acid (puriss. p.a. for mass spectroscopy), methanol (p.a. ACS), acetonitrile (LC-MS grade), and acetic acid (puriss.) were purchased from Sigma–Aldrich. Ultrapure water was obtained from a Milli-Q UF Plus water system (Millipore, Molsheim, France).

### 4.8. RNA Isolation and RT-qPCR Analysis

Cells were washed twice with PBS and harvested in the cell lysis buffer provided with the NucleoSpin RNA II Purification Kit (Macherey Nagel, Düren, Germany). Total RNA was then isolated according to the manufacturer’s instructions. The mRNA levels were determined by RT-qPCR. The primers were designed to flank the exon-exon junctions of transcripts for amplification of cDNA only. For the full list of the primer sequences (designed and synthesized by Generi Biotech, Czech Rep.) with the numbers of Universal probe Library probes (UPL; Roche Life Sciences, Mannheim, Germany) or Taqman^®^ gene expression assays (Thermo Fisher Scientific, Waltham, MA, USA) for all detected genes, see the Appendix A Appendix A. In all experiments, the average of the values from the three reference genes was used—HMBS (NM_000190), POLR2A (NM_000937; both Generi Biotech, #3033-F), and IPO8 (Taqman^®^ assay #Hs00183533_m1). The amplifications were carried out in 20 µL of reaction mixture containing: 10 µL of QuantiTect Probe RT-PCR Master Mix, 0.2 µL of QuantiTect RT mix (Qiagen GmbH, Hilden, Germany), 2 µL of primers and probe solution, 5.8 µL of water, and 2 µL of sample and were run on the LightCycler (Roche) using the following program: reverse transcription at 50 °C for 20 min and an initial activation step at 95 °C for 15 min, followed by 40 cycles at 95 °C for 15 s, and 60 °C for 60 s. Changes in gene expression were calculated using the comparative threshold cycle method [48].

### 4.9. Statistical Analysis

All data in this study were analyzed by GraphPad Prism v7 software using normality tests (D’Agostino and Pearson or Shapiro-Wilk) followed by unpaired *t*-test, Mann Whitney test or by multiple *t*-test (Benjamini, Hochberg, Yekutieli) if not specified otherwise. Differences were considered significant when *p* < 0.05 or *q* < 0.05, respectively.

## Figures and Tables

**Figure 1 ijms-22-09195-f001:**
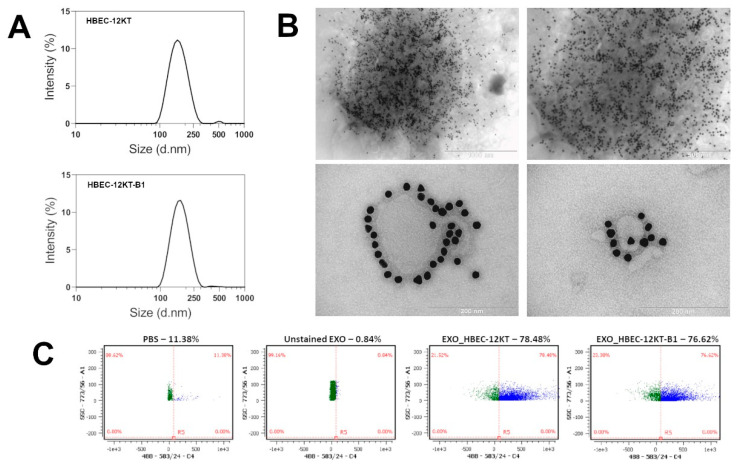
Characterization of isolated exosomes. (**A**) The size distribution of the exosomes isolated from HBEC-12KT and HBEC-12KT-B1 conditioned media was analyzed using multi-angled dynamic light scattering on Zetasizer Ultra. The representative results of the size distribution analysis (n = 5) are displayed. (**B**) The isolated exosomes were labeled with specific primary anti-CD63 and gold particle conjugated secondary antibodies and visualized by TEM. The representative images show clusters of labeled exosomes (top) and detailed visualization of labeled exosome released from HBEC-12KT (bottom left) or HBEC-12KT-B1 (bottom right) cells. (**C**) Flow cytometric analysis (CellStream, Luminex) of the isolated exosomes stained (or unstained) with PE-conjugated specific CD63 antibody. The representative results (n = 5) are shown.

**Figure 2 ijms-22-09195-f002:**
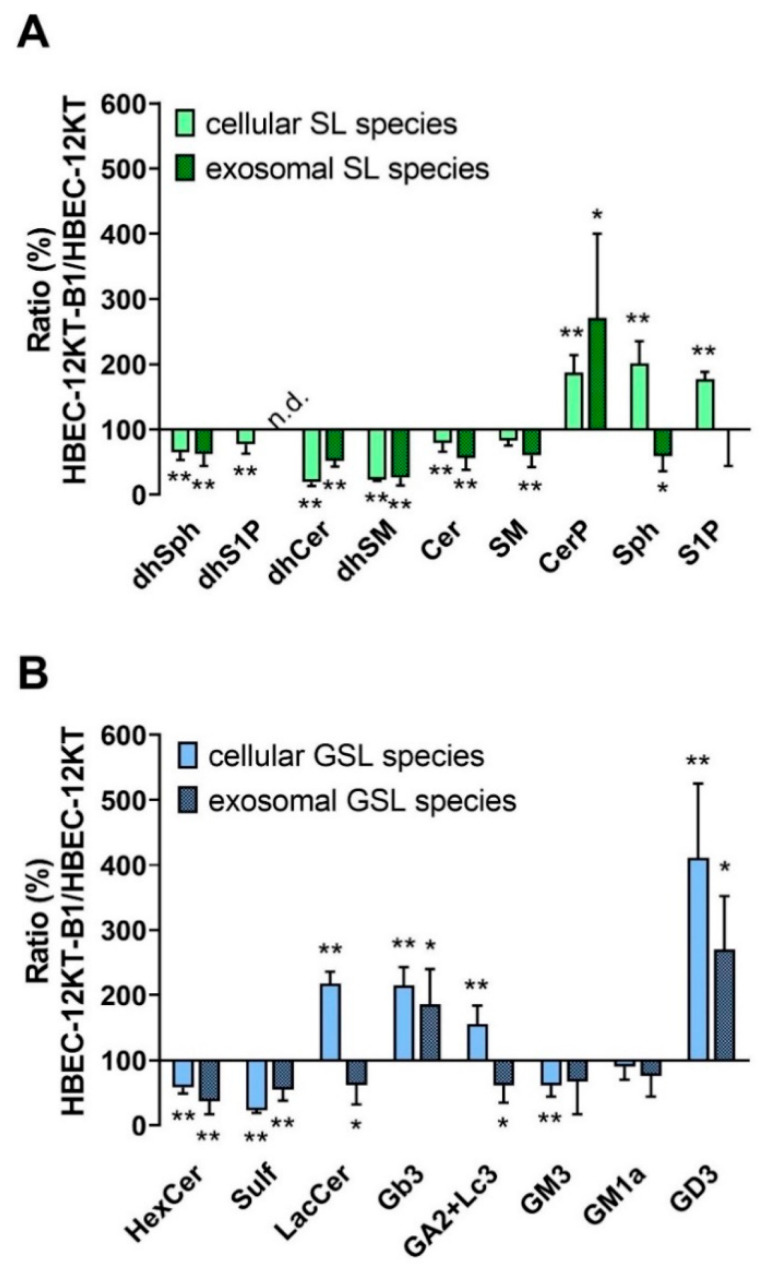
Relative concentration of SL and GSL species in BaP-transformed HBEC-12KT-B1 cells vs. parental HBEC-12KT cells, as compared with their content in isolated exosomes. (**A**) Sum of SL species in cells and exosomes; (**B**) sum of GSL species. The cells and isolated exosomes were fixed in methanol, extracted, and analyzed with LC-MS/MS, as described in the material and methods. The results represent mean values ± S.D. of three independent experiments performed in duplicates (the cells) or the mean values ± S.D. of five independent experiments performed in monoplicates (exosomes). They are expressed as a ratio (in %) of values obtained in transformed (HBEC-12KT-B1) vs. non-transformed (HBEC-12KT) cells. The symbols ‘*’ and ‘**’ denote significant difference (*p* < 0.05 and *p* < 0.01, respectively).

**Figure 3 ijms-22-09195-f003:**
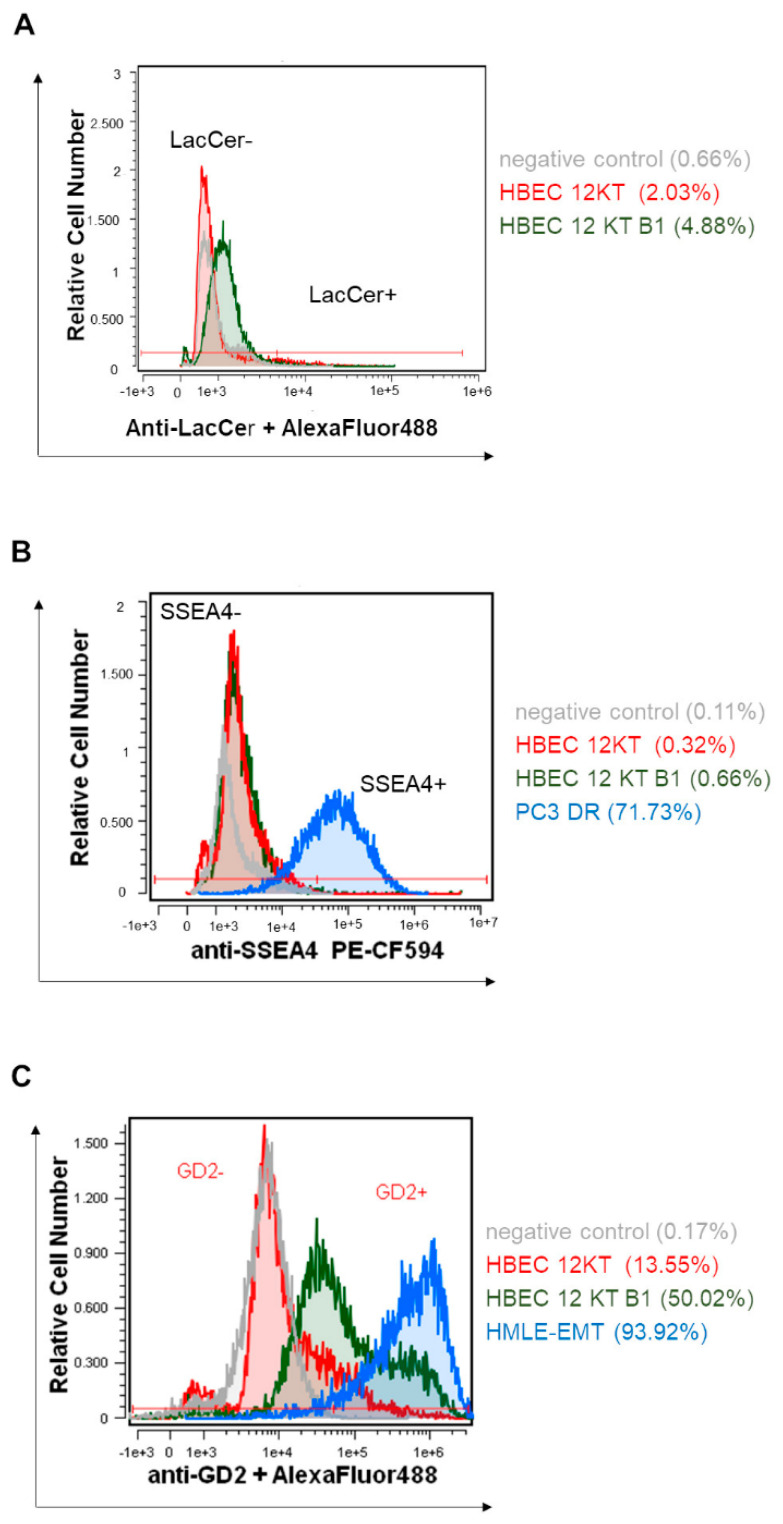
Surface levels of LacCer, SSEA-4, and GD2 on HBEC-12KT-B1 and HBEC-12KT cells. Representative overlays of flow cytometric histograms show cells stained with mouse anti-human CDw17/LacCer and anti-mouse IgG/IgM AlexaFluor 488 antibodies (**A**); SSEA4 PE-CF594 antibody (**B**); and with mouse anti-human GD2 and anti-mouse IgG Alexa Fluor 488 antibodies (**C**). PC3 DR cells and HMLE-EMT cells were used as positive controls for SSEA4 and GD2, respectively. Negative control (gray), HBEC 12 KT (red), HBEC 12 KT B1 (green), positive controls (blue). The data represent mean values ± standard deviation (n = 3).

**Figure 4 ijms-22-09195-f004:**
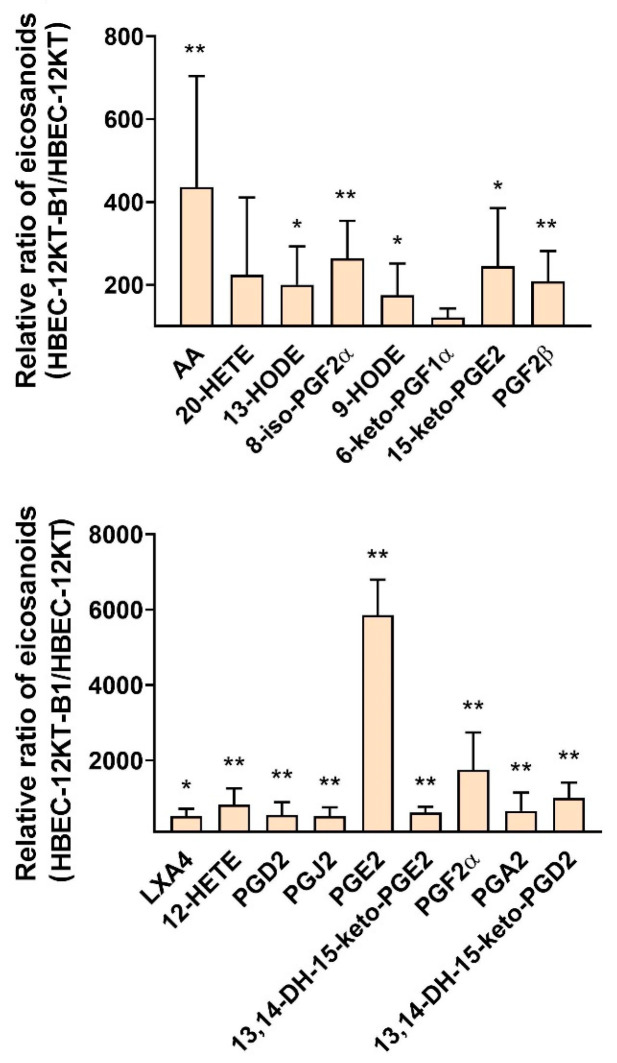
(**A**,**B**) Comparison of concentrations of eicosanoids released from transformed HBEC-12KT-B1 and non-transformed HBEC-12KT cells. AA and the evaluated eicosanoids were isolated from cell culture media by solid-phase extraction and then analyzed by HPLC-MS/MS. The results represent means ± standard deviations obtained in three independent experiments in triplicates. The symbols ‘*’ and ‘**’ denote a significant difference (*p* < 0.05 and *p* < 0.01, respectively). Abbreviations: AA, arachidonic acid; HETE, hydroxyeicosatetraenoic acid; HODE, hydroxyoctadecadienoic acid; PG, prostaglandin.

**Figure 5 ijms-22-09195-f005:**
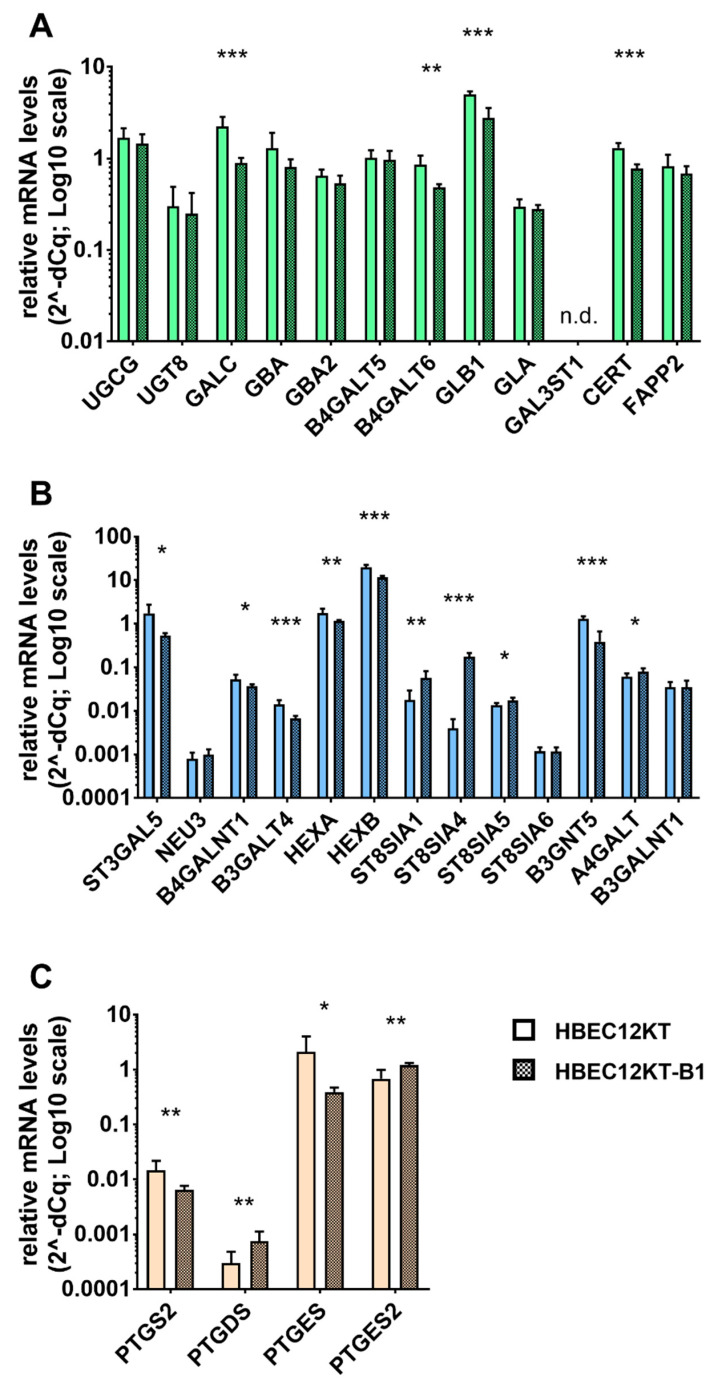
Changes in the expression of genes encoding enzymes of lipid metabolism in non-transformed human bronchial epithelial HBEC-12KT cells and their transformed counterparts, HBEC-12KT-B1 cells. Relative mRNA levels of genes encoding enzymes of HexCer and LacCer metabolism (**A**), GSL metabolism (**B**), and the metabolism of eicosanoids (**C**) were detected using RT-qPCR as described in the materials and methods. The asterisks indicate a significant difference in the given mRNA level between epithelial HBEC-12KT cells and HBEC-12KT-B1 (patterned bars) as determined by the multiple *t*-test; ‘*’ *q* < 0.05, ‘**’ *q* < 0.01, ‘***’ *q* < 0.001.

**Figure 6 ijms-22-09195-f006:**
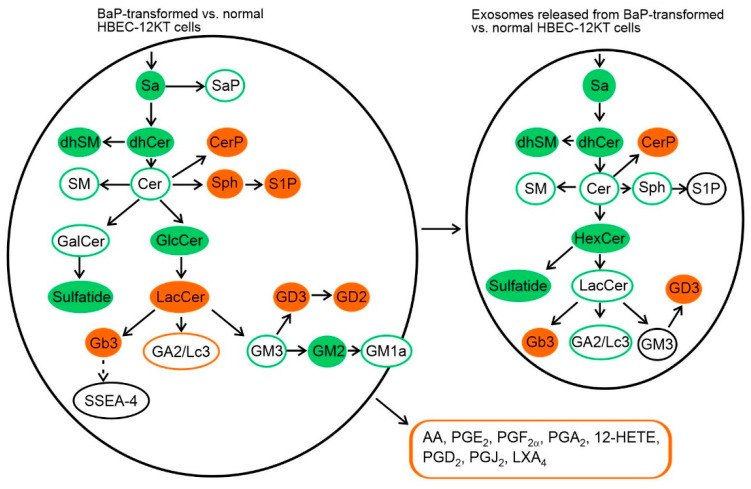
Summary of relative changes in the SL and GSL profiles in the BaP-transformed HBEC-12KT-B1 cells as compared with the non-transformed human bronchial epithelial HBEC-12KT cells. Red: increased levels; green: decreased level. The increased release of eicosanoids from the transformed cells is indicated in a separate box.

**Table 1 ijms-22-09195-t001:** Size, polydispersity (PdI) and concentrations of the isolated exosomes, and numbers of cells used for isolation of exosomes. All values represent means ± standard deviation (n = 3).

Cell Line	Z-Average (d.nm)	PdI	Number of Particles per mL	Number of Particles per Sample	Number of Cells per Sample	Number of Particles per Million Cells
HBEC-12KT	158 ± 7	0.130 ± 0.009	5.58 ± 2.46 × 10^11^	2.51 ± 1.11 × 10^10^	1.01 ± 0.31 × 10^8^	2.53 ± 1.23 × 10^8^
HBEC-12KT-B1	162 ± 8	0.123 ± 0.030	8.55 ± 2.08 × 10^11^	3.85 ± 0.94 × 10^10^	1.85 ± 0.38 × 10^8^	2.26 ± 0.70 × 10^8^

## Data Availability

Original data are available upon reasonable request to corresponding author.

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
