# Peer review of "Changes in Sphingolipid Profile of Benzo[a]pyrene-Transformed Human Bronchial Epithelial Cells Are Reflected in the Altered Composition of Sphingolipids in Their Exosomes"

_ijms, 2021, doi:10.3390/ijms22179195_

Round 1
Reviewer 1 Report
The work from Miroslav Machala and co-worker is very interesting. The investigation of the lipid profile of exosomes in comparison to the cellular lipid profile is well done and contribute to a better understanding of the communication mechanisms between cells.
However I have some remarks to this manuscript that might help to improve it.
1) The title might be more expressive if the mean message of this manuscript is clearly included. For example: “Changes in the lipid profile of benzo[a]pyrene transformed human bronchial epithelial cells is reflected in the lipidomic profile of their exosomes”
2) The eicosanoid data are a nice add-on to the sphingolipid data, but their interconnections to the sphingolipid changes are not proven. However their interrelationship are discussed, but is there any hint that changes in the sphingolipid profile is related to changes in the eicosanoid level or vice versa?
Furthermore, the eicosanoids are measured in the supernatant of the cells, because they are released from cells as soluble mediators into the media. However, it should be elaborate that these metabolites are not measured in exosomes. Or did you measure the eicosanoid level also in exosomes?
At least it is astonishing that although PGE2 increased about 6000-fold in HBEC-12KT-B1 cells in comparison to the non-transformed HBEC-12 KT cells but neither COX-2 nor mPGES1 expression increased. To verify these contradictory data the expression level of COX-2 and mPGES1 might be shown by Western-blot analysis. At least these data should be addressed in the discussion.
3) Even the authors relegate the readers to their publication (here the citation is not transformed into a number in the text, Page 14, line 496) where the primers for RT-PCR are already published, I think it is an essential information which should be directly given in this manuscript.
Author Response
Response to reviewers
Reviewer no. 1
We would like to thank the reviewer for the very positive comments on our manuscript and valuable comments. We respond to the individual comments below:
General Comments to the Author
The work from Miroslav Machala and co-worker is very interesting. The investigation of the lipid profile of exosomes in comparison to the cellular lipid profile is well done and contribute to a better understanding of the communication mechanisms between cells.
However I have some remarks to this manuscript that might help to improve it.
Individual comments:
Comment 1: The title might be more expressive if the mean message of this manuscript is clearly included. For example: “Changes in the lipid profile of benzo[a]pyrene transformed human bronchial epithelial cells is reflected in the lipidomic profile of their exosomes”
Answer to Comment 1:
We modified the title of the manuscript according to reviewer´s suggestion: „ Changes in sphingolipid profile of ben-zo[a]pyrene-transformed human bronchial epithelial cells are reflected in the altered composition of sphingolipids in their exosomes“.
Comment 2: The eicosanoid data are a nice add-on to the sphingolipid data, but their interconnections to the sphingolipid changes are not proven. However their interrelationship are discussed, but is there any hint that changes in the sphingolipid profile is related to changes in the eicosanoid level or vice versa?
Furthermore, the eicosanoids are measured in the supernatant of the cells, because they are released from cells as soluble mediators into the media. However, it should be elaborate that these metabolites are not measured in exosomes. Or did you measure the eicosanoid level also in exosomes?
At least it is astonishing that although PGE2 increased about 6000-fold in HBEC-12KT-B1 cells in comparison to the non-transformed HBEC-12 KT cells but neither COX-2 nor mPGES1 expression increased. To verify these contradictory data the expression level of COX-2 and mPGES1 might be shown by Western-blot analysis. At least these data should be addressed in the discussion.
Answer to Comment 2:
Thank you for this comment. We added more information about relationships of changed SL pattern and altered eicosanoid synthesis, including the respective references in Introduction (p. 2, line 80), Results (p. 5, line180) and Discussion (p. 9, line 285):
“A direct link has been identified between aberrant levels of phosphorylated SLs and activation of cytosolic phospholipase 2Aalpha, induction of arachidonic production, which is the rate-limiting step in eicosanoid synthesis, as well as activation of cyclooxygenase-2, as the first enzyme of prostaglandin biosynthesis [Pettus et al. Curr. Mol. Medicine, 2004, Kawamori et al. FASEB j., 2009].“
“Induction of CerP and S1P synthesis might be involved might be involved also in induction of eicosanoid metabolism...“
“Induction of both CerP and S1P have been also implicated in activation of arachidonic acid metabolism...“
Regarding the second part of this point, in our study, we opted to determine total levels of released eicosanoids, as they are directly modulating cancer cell behavior in both autocrine and paracrine fashion. Nevertheless, the suggestion to measure eicosanoids also directly in isolated exosomes is valuable and we intend to perform it in our future work.
In addition, we extended discussion on modulation of expression of genes linked to PGE2 metabolism (p. 10, line 301):
“Nevertheless, it seems that activation of cytosolic PLA2alpha and PTGS2/COX-2 and not changes in their expression plays a major role in induction of eicosanoid metabolism. This is in accordance with previous studies showing that increase in CerP and S1P and decrease in SM intracellular concentrations (which was the altered SL pattern found in mesenchymal HBEC-12KT-B1 cells in our study) have been implicated in the induction of eicosanoid metabolism......“
Comment 3: Even the authors relegate the readers to their publication (here the citation is not transformed into a number in the text, Page 14, line 496) where the primers for RT-PCR are already published, I think it is an essential information which should be directly given in this manuscript.
Answer to Comment 3:
We prepared new Supplementary Table 3 that provides the information about the primers/probes used in our RT-PCR analyses.
Reviewer 2 Report
This manuscript describes the contents of examining the lipid profile of cells and exosomes using transformed bronchial epithelial cells by BaP. I have a brief comment to the author.
Comment 1. This is an experimental system based on Ref. 28, and I think it is composed of appropriate techniques and stories. Elevated levels of ceramide phosphate, globoside Gb3, and ganglioside GD3 in transformed cell-derived exosomes are considered positive findings, but cellular and exosomal kinetics did not match for Sph, S1P, and LacCer. I couldn't quite understand what it means and what it does when placed in the microenvironment of transformed cells and cancer.
Comment 2. Is the concentration and duration of BaP (1 μM) for 12 weeks a sufficient setting as a model of BaP exposure to clinical bronchial epithelial cells?
Author Response
Response to reviewers
Reviewer no. 2
Authors would like to thank the reviewer for the positive evaluation of our manuscript and valuable comments. We respond to the individual comments below:
General Comments to the Author
This manuscript describes the contents of examining the lipid profile of cells and exosomes using transformed bronchial epithelial cells by BaP. I have a brief comment to the author.
Comment 1: This is an experimental system based on Ref. 28, and I think it is composed of appropriate techniques and stories. Elevated levels of ceramide phosphate, globoside Gb3, and ganglioside GD3 in transformed cell-derived exosomes are considered positive findings, but cellular and exosomal kinetics did not match for Sph, S1P, and LacCer. I couldn't quite understand what it means and what it does when placed in the microenvironment of transformed cells and cancer.
Answer to Comment 1:
Based on our HPLC-MS/MS data, it really seems that Sph, S1P and Lac Cer content in exosomes did not reflect the SL/GSL pattern in the cells. The implication would be that the exosomes, therefore, cannot carry the increased levels of these specific SLs/GSLs (which have been also shown to contribute to cancer progression) and, consequently, they are not likely to affect the recipient cells by these altered SL patterns. The individual mechanism(s) underlying the selective enrichment of some SLs/GSLs in exosomes and their levels in tumor microenvironment should be explored in future studies. We also included this sentence in Discussion (p. 11, line 372): “The results obtained in this study should be further explored in future experiments, including the mechanism(s) linked to relatively lower content of Sph, S1P and LacCer found in exosomes derived from transformed cells.“
Comment 2: Is the concentration and duration of BaP (1 μM) for 12 weeks a sufficient setting as a model of BaP exposure to clinical bronchial epithelial cells?
Answer to Comment 2:
The BaP concentration that was used for cellular transformation was sufficient to induce morphological and functional changes reflecting the transformation of the cells, including epithelial-mesenchymal transition, the ability to grow in soft agar, stimulation of cell migration etc. (please see Bersaas et al., 2016, for further details – the reference is included in our manuscript). We also provide extended information about the transformation of cells in the revised manuscript (Discussion, p. 10, line 327): “In our study, we used human HBEC cells transformed with chemical carcinogen – BaP. The cell transformation of HBEC-12KT has led to both morphological and functional changes including enhanced migration, the ability to form colonies in soft agar, as well as to induction of mesenchymal phenotype reflecting EMT [30].” Therefore, this is highly relevant and stable cellular model suitable for comparison of lipidome in parental normal, non-carcinogenic, epithelial cells and their mesenchymal counterpart.